# Exploring the Use of Antibiotics for Dental Patients in a Middle-Income Country: Interviews with Clinicians in Two Ghanaian Hospitals

**DOI:** 10.3390/antibiotics11081081

**Published:** 2022-08-09

**Authors:** Jacqueline Sneddon, Wendy Thompson, Lily N. A. Kpobi, Diana Abena Ade, Israel Abebrese Sefah, Daniel Afriyie, Joanna Goldthorpe, Rebecca Turner, Saher Nawaz, Shona Wilson, Jo Hart, Lucie Byrne-Davis

**Affiliations:** 1Healthcare Improvement Scotland, Glasgow G1 2NP, UK; 2British Society for Antimicrobial Chemotherapy, Birmingham B1 3NJ, UK; 3Division of Dentistry, University of Manchester, Manchester M13 9PL, UK; 4Regional Institute for Population Studies, University of Ghana, Legon, Accra P.O. Box LG25, Ghana; 5Pharmacy Practice Department, School of Pharmacy, University of Health and Allied Sciences, Ho P.O. Box PMB31, Ghana; 6Keta Municipal Hospital, Keta-Dzelukope P.O. Box WT82, Ghana; 7Ghana Police Hospital, Accra P.O. Box CT104, Ghana; 8Division of Medical Education, University of Manchester, Manchester M13 9PL, UK

**Keywords:** prescribing, antimicrobial resistance, dental infection

## Abstract

Background: Antimicrobial resistance is a global problem driven by the overuse of antibiotics. Dentists are responsible for about 10% of antibiotics usage across healthcare worldwide. Factors influencing dental antibiotic prescribing are numerous, with some differences in low- and middle-income countries compared with high-income countries. This study aimed to explore the antibiotic prescribing behaviour and knowledge of teams treating dental patients in two Ghanaian hospitals. Methods: Qualitative interviews were undertaken with dentists, pharmacists, and other healthcare team members at two hospitals in urban and rural locations. Thematic and behaviour analyses using the Actor, Action, Context, Target, Time framework were undertaken. Results: Knowledge about ‘antimicrobial resistance and antibiotic stewardship’ and ‘people and places’ were identified themes. Influences on dental prescribing decisions related to the organisational context (such as the hierarchical influence of colleagues and availability of specific antibiotics in the hospital setting), clinical issues (such as therapeutic versus prophylactic indications and availability of sterile dental instruments), and patient issues such as hygiene in the home environment, delays in seeking professional help, ability to access antibiotics in the community without a prescription and patient’s ability to pay for the complete prescription. Conclusions: This work provides new evidence on behavioural factors influencing dental antibiotic prescribing, including resource constraints which affect the availability of certain antibiotics and diagnostic tests. Further research is required to fully understand their influence and inform the development of new approaches to optimising antibiotic use by dentists in Ghana and potentially other low- and middle-income countries.

## 1. Introduction

Antimicrobial resistance (AMR) is a global problem that poses a significant threat to health and wealth, due to prolonged illnesses, longer hospital stays, and increased mortality [1]. The World Bank has identified that AMR could push an additional 28.3 million people into extreme poverty by 2050, most of them living in low- and middle-income countries (LMICs) [2]. The World Health Organization (WHO) has highlighted the urgency of tackling AMR [3]. The WHO global action plan (GAP) on AMR aims ‘to ensure, for as long as possible, continuity of the ability to treat and prevent infectious diseases with effective and safe medicines’ [3]. Three of the WHO’s objectives for implementation have been identified by Fédération Dentaire Internationale (FDI) World Dental Federation as pertinent to dental teams: awareness-raising, preventing infections, and antibiotic stewardship (ABS) [4]. ABS is a coherent set of actions which promote the appropriate use of antibiotics, in ways that ensure sustainable access to effective therapy for all who need them [5].

Dentists are responsible for about 10% of antibiotic prescribing for humans worldwide, with a high rate of unnecessary and inappropriate use [5]. Patterns of dental antibiotic prescribing are known to differ between countries [6] influenced by a range of factors, including differences in clinical guidelines (such as therapeutic versus prophylactic indications), clinician beliefs, access to dental care, and whether practiced in high-income or low- and middle-income countries [4,7]. A general misunderstanding amongst the public exists indicating that antibiotics are required to treat toothache and also that their use can avoid surgical treatment such as extraction of the tooth [8]. However, dental pain is often caused by an inflammation of the pulp, and infections are usually amenable to treatment through a procedure rather than antibiotics [9,10]. Accurate diagnosis is required to ensure appropriate treatment is provided, and diagnosis usually requires a dental X-ray. Unlike in conditions managed by medical practitioners, culture and sensitivity testing is rarely appropriate for dental infections.

There is no one-size-fits-all approach to dental antibiotic stewardship (ABS), [4], and it is important to understand the reasons (clinical and non-clinical) why antibiotics are prescribed for dental conditions in a relevant context, before designing solutions to optimise antibiotic use in that context. Several successful dental ABS interventions have been reported in the literature, with just one administered in an LMIC: an educational feedback intervention in Nepal [11].

Dentists in LMICs work mainly in hospitals, where primary dental care is provided to outpatients [7]. Ghana is a middle-income country, and the two hospitals involved in this study have delivered successful AMS programmes [12]. Whilst dentistry was not included in either of these AMS programmes, recent audits in both hospitals found high rates of dental antibiotic use: dental teams were the highest prescribers of antibiotics, responsible for >20% of antibiotics prescribed in one hospital [13]. The audits also found high rates of antibiotic misuse: nearly 90% of dental patients in an-other hospital received an antibiotic, of which 88% were non-compliant with national standard treatment guidelines [14]. This study aimed to explore the antibiotic prescribing behaviour of dental teams and knowledge/beliefs about antibiotic stewardship at these two hospitals in Ghana.

## 2. Method

Qualitative interviews were undertaken with hospital staff involved in the prescription and supply of antibiotics for patients presenting with dental infections.

### 2.1. Study Setting

Two hospitals in Ghana were the setting for this study: One is a 100-bed hospital in Accra (Ghana’s capital city), and the other is a 110-bed government hospital in a rural area within the Volta Region. These hospitals were selected due to their involvement in a previous AMS project and previous studies of dental antibiotic use [13,14].

### 2.2. Participants

Staff from each of the following groups were recruited for the study:Dental and nursing staff who prescribe antibiotics for people with dental conditions in outpatient departments;Staff who prescribe antibiotics for dental inpatients;Pharmacy staff who dispense antibiotics and review prescriptions for antibiotics in dental patients.

Recruitment took place during July and August 2021. The lead pharmacist at each hospital identified staff and made an initial approach in person, offering the participant information sheet (Appendix A) and giving potential participants at least 48 h to decide whether to participate and an opportunity to ask questions about the study. Those who agreed to participate were asked to sign a consent form after which their contact details were shared with the interviewers.

### 2.3. Data Collection

A semi-structured interview schedule and a topic guide (Appendix A) were developed, based on previous research [4], and piloted with two members of the dental teams at each hospital. Based on this previous study, a sample size of 15 interviews across the two hospitals was selected as sufficient to achieve saturation of the themes.

Interviews were conducted either face to face or via telephone by a non-clinical researcher (DAA) from the University of Ghana (UoG) between August and October 2021. Participants completing the interview were remunerated for their time with the provision of lunch or purchase of mobile data to the value of GHS 41 (approximately £5). Interviews were audio-recorded and transcribed (including anonymisation and depersonalisation) by a professional transcription service based in Ghana, using an intelligent verbatim approach. All transcripts were quality-controlled by the work of the original transcriptionist being independently checked by one of the researchers (L.N.A.K.). Data were stored securely in UoG systems during the period of the project and accessible only to the research team. All data were deleted once the study was complete.

### 2.4. Data Analysis

Interview transcripts were thematically analysed by the research team (as per Braun and Clarke [15]), including reading backwards and forwards through each transcript (principle of constant comparison) [16]. The researchers looked for evidence for and against each theme, and results were presented by theme, using illustrative quotes where appropriate. Additionally, interviews were analysed for specific behaviours related to the use of antibiotics. These were specified using the Actor, Action, Context, Target, Time (AACTT) framework [17].

### 2.5. Ethics Approval

Ethical approval for the qualitative elements was obtained from the Ghana Health Service Ethics Review Committee before any data were collected (reference GHS-ERC-5 May 2021). All researchers were trained in research ethics. Permission to conduct the research was obtained from senior management at each hospital.

## 3. Results

For this study, 12 interviews were conducted (4 dentists, 4 pharmacists, 2 pharmacy technicians, 1 dental surgery assistant, and 1 physician assistant), 7 at one hospital and 5 at the other.

### 3.1. Behavioural Specification

Multiple behaviours were identified across professional groups, which were specified using the AACTT framework (see Table 1).

### 3.2. Key Themes and Subthemes

Two key themes, namely (1) people and places and (2) knowledge and beliefs about AMR and ABS, and ten subthemes were identified; these are presented with illustrative quotes in Table 2.

## 4. People and Places

### 4.1. Prescriber and Clinical Context

Members of the dental team and physician prescribers talked about the national standard treatment guideline providing advice on appropriate antibiotic prescribing. Some also talked, however, about their personal preferences (P drug). Quotes from participants on the theme of ‘people and places’ are shown in Table 3.

#### 4.1.1. Prescribing Practices

Antibiotics were viewed as a fundamental aspect of healthcare and were prescribed on a daily basis for a variety of therapeutic and prophylactic indications within the hospital dental clinics. Amoxicillin and metronidazole were the antibiotics mentioned most often by interviewees, although some of the dentists mentioned broad-spectrum drugs such as co-amoxiclav as their first line for dental infections.

##### Therapeutic Use for Treating Active Infections

Antibiotic prescriptions were commonly used to treat patients with suspected dental infections. The aim of antibiotic prescribing was to relieve pain and treat active infections.

Dentists most frequently reported prescribing for localised and spreading dental infections such as dentoalveolar abscesses and Ludwig’s angina. However, there was a mismatch with what members of the pharmacy team reported as being the diagnoses for which antibiotics were most often prescribed by the dental team; conditions related to the microbiome (such as caries and periodontal disease) were identified by the pharmacy teams rather than frank infections such as periapical abscesses.

##### Prophylactic Use to Prevent Infections

Often, antibiotics were used in dentistry as a preventative measure, to protect against complications from possible local and distance site infections, in patients with underlying health conditions (e.g., those at risk of infective endocarditis).

#### 4.1.2. Resource Constraints

Having access to appropriate resources to deliver care in accordance with standard treatment guidelines was identified as an important influence on dental antibiotic prescribing.

##### Access to Antibiotics—Availability and Affordability

Some interviewees described occasions on which particular antibiotics were unavailable within the hospital.

However, the pharmacists highlighted that unavailability is unusual for dental patients, as the antibiotics most commonly used are readily available in Ghana (P drugs).

Due to the nature of the healthcare system in Ghana, some patients had restricted access to care if they were not insured.

Dentists’ perceptions about patients’ abilities to pay for antibiotics influenced the decision of which antibiotic was prescribed, even if the clinician was unsure about the efficacy of the selected drug for the condition being treated.

##### Diagnostic Testing

The results of diagnostic tests (such as dental radiographs and culture and sensitivity testing) were discussed by some participants as influencing whether or what (e.g., narrow-spectrum drugs) to prescribe (respectively).

When there was access to such laboratory support, decision making on prescriptions could change in light of the results, with prescribing sometimes then not in line with guidance.

##### Infection Prevention and Control

Some members of the dental team expressed concern about the cleanliness of the facilities and the sterility of instruments within their clinics.

Extensive use of antibiotics to avoid post-operative infections following dental surgery, such as removal of teeth, was reported.

#### 4.1.3. Influence of colleagues—Hierarchies

Senior colleagues influenced junior colleagues’ decision making in antibiotic prescribing. The most knowledgeable and experienced dentist usually made the decisions or aided junior dentists’ decision making. Some expressed concern that this would lead to passing on habitual prescribing patterns, which was concerning if not in line with current guidance.

### 4.2. Pharmacists/Dispensers

Whilst members of the pharmacy team did not prescribe antibiotics in the hospital setting, many reported that they did when working in the community setting.

Members of the pharmacy team were used to receiving visits from pharmaceutical company representatives but felt this did not change their practice.

#### Availability of Antibiotics without a Prescription

There was a general perception that if participants were not prescribed antibiotics, then patients could still access antibiotics within the community without a prescription from community pharmacies or chemical sellers.

Where poor access to dentistry existed, patients were felt to be particularly likely to source antibiotics before seeking dental care.

### 4.3. Patients

#### 4.3.1. Delays in Seeking Professional Help

Commonly, patients delayed treatment and presented at the dentist when intervention was required; participants described this as part of their ‘culture’ for individuals to delay seeking help.

Some clinicians described patient expectations and demands for antibiotics when presenting for professional care.

#### 4.3.2. Hygiene Concerns

Antibiotics were prescribed as prophylaxis following dental procedures even if there was no infection present prior to surgery, due to the clinicians’ concerns about suboptimal home environments or poor oral hygiene.

#### 4.3.3. Medication Adherence

Commonly, patients were perceived to not adhere to their antibiotic prescriptions due to not being able to afford the complete prescription.

## 5. Knowledge and Training about AMR and AMS

Variable levels of knowledge and training on AMR and AMS were reported. Again, a hierarchical issue was identified, with senior staff receiving training but not the wider clinical team. Quotes from participants on theme of ‘knowledge and training about AMR and AMS’ are shown in Table 4.

### 5.1. Consequences of AMR

Participants generally had good knowledge of AMR and how resistance was a ‘global issue’ within dentistry and across different fields of medicine. However, this did not seem to have a significant influence on the decision making around antimicrobial prescribing, as antibiotics were often prescribed prophylactically and to treat dental conditions caused by inflammation and/or the oral microbiome (such as tooth decay (caries), gum disease (periodontitis) or toothache (pulpitis)) rather than a bacterial infection.

AMS was perceived as an important intervention, viewed as a way to tackle the growing threat of AMR.

### 5.2. Causes of AMR

Participants’ knowledge about the causes of AMR was inconsistent. Many felt it was predominantly the misuse of antibiotics by patients.

There was less awareness about the role healthcare professionals play in AMR.

The availability of antibiotics within the community without prescriptions was perceived as driving the global issue of AMR.

### 5.3. Role of Prescribers in Tackling AMR

Counselling and discussing antibiotic use and prescription with patients were seen as part of the role of prescribers in helping to tackle AMR.

## 6. Discussion

A range of prescriber, dispenser, and patient factors were identified as important influences on antibiotic prescribing for dental patients in Ghana, a middle-income country, with limited provision for dental services across the whole community. Despite the awareness of antimicrobial resistance and the importance of following good clinical practice, widespread prophylactic use of antibiotics was described, and prescribers were following the advice of senior colleagues rather than national standard treatment guidelines.

Many of the issues have also been identified by previous studies across both high-income countries and LMICs, including patient expectations for antibiotics and awareness about antimicrobial resistance [7,18]. Some identified factors had previously been identified as unique to LMICs, such as the influence of poor oral hygiene of the patient, affordability of antibiotics to patients, and access to antibiotics in the community without a prescription [18,19,20]. The hierarchical influence of senior prescribers on junior colleagues is well-known within the UK hospital setting but has not previously been reported in dental studies undertaken in community settings [21]. The impact of unsanitary living conditions, unsterile dental instruments, and the restricted availability of some types of antibiotics in hospitals were factors which have not previously been reported in the dental literature.

Prescribers considered antibiotics appropriate to relieve pain, treat active infections, and as an adjunct to prevent local and distance-site infections in patients undergoing a dental procedure, such as extraction of a tooth. The use of antibiotics to treat dental pain in the absence of infection has been widely reported in studies of dental antibiotic prescribing across a wide range of countries [9,22,23,24,25]. In these countries, further research has been undertaken to understand the reasons why clinicians treat dental conditions caused by inflammation but not infection with antibiotics. Reasons have included diagnostic difficulties, habit, beliefs about the dentist’s own ability to treatment, acute dental pain during an urgent dental appointment, and workload/time factors. A lack of access to diagnostic testing to guide treatment was suggested as a reason for widespread antibiotic prescribing for dental patients and the use of broad-spectrum agents. As highlighted in the recently published World Health Organization’s essential medicines antibiotics handbook [26]. Dental radiographs (X-rays) are important for diagnosing dental infections, but empirical prescribing is usually indicated due to the polymicrobial nature of dental infections and the speed with which dental infections can spread [27]. Whilst this study made no assessment of the Ghana standard treatment guidelines, it is notable that, in Nigeria, the standard treatment guidelines recommend oral systemic antibiotics for inflammatory conditions such as gingivitis and periodontitis [28], whereas the European Federation of Periodontal Guidelines states this is not an appropriate indication [29]. Further research is indicated to review the Ghana standard treatment guidelines to ensure they align with the latest international evidence-based guidelines.

As reported in this study, many believe that failure to complete a course of antibiotics is a key driver of antibiotic resistance. The concept of a course of antibiotics has been challenged in the literature [30] and in some countries, the guidelines for dental antibiotic prescribing now advise stopping the patient from taking antibiotics once the symptoms have resolved [31]. The use of antibiotics in the community without a prescription is also reported in other studies of dental antibiotic prescribing in LMICs, [32] with some of the participants keen to highlight that they would advise patients against this practice. As per the FDI World Dental Federation white paper on antibiotic resistance, spreading awareness about antibiotic resistance in this way is an important role which all members of the dental team can play in tackling this global problem [4]. Interestingly, pressure from pharmaceutical companies to prescribe in certain ways has also been reported in several studies undertaken in LMICs across both dental and medical settings [7,32], but this was not a finding of this study.

This study confirmed that there are factors influencing dental antibiotic prescribing in Ghanaian hospitals which are not addressed in any current ABS programmes around the world. As highlighted by the range of influences on dental antibiotic prescribing identified, this is a complex behaviour for which a complex intervention [33] targeting various behavioural influences is most likely to successfully reduce unnecessary antibiotics [34]. Addressing clinicians’ capabilities by providing guidelines and improving their knowledge are known to be necessary but not sufficient to change their behaviour [35]. Using the principles of the behaviour change wheel [34], changing the social environment in which antibiotics are prescribed as part of hospital-based dental care might include working with the hierarchical structure to ensure senior clinicians are engaged with best practices. This may involve pressing for a review and update of national standard treatment guidelines or collaborative working between senior dentists and pharmacists to develop a local guideline. An education programme to support the implementation of guidance along with actions to tackle patient demands and expectations are required to achieve success. This might involve behaviourally informed public awareness campaigns about the futility of antibiotics for toothache and the need to seek early help for dental conditions to avoid the potential danger of inappropriate antibiotic use. However, the impact of patient education through posters and leaflets should be carefully assessed, as they can have unintended consequences [36,37]. Tackling motivational factors such as dentists’ beliefs about the therapeutic and prophylactic benefits versus the risks of adverse outcomes due to antibiotics could also be important.

A key limitation of this study is that it was based on interviews with hospital-based dental and pharmacy teams. Studies relying on self-reported perceptions are inherently at risk of bias in the way people account for their own behaviour and that of others, as people are often unaware of what influences their unconscious/instinctive behaviour in practice [38]. The next steps are to gain a better understanding of the factors influencing decision making when treating dental patients with (and without) antibiotics across sub-Saharan Africa, including the culture and beliefs of dentists and patients, and community perspectives. A further study is planned to enable in-depth understanding and application of behavioural science to support the development of tailored interventions for optimising dental antibiotic prescribing. This study will encompass, both prophylactic prescribing to prevent local and distant site infections following dental procedures and therapeutic prescribing for oral and dental infections.

## 7. Conclusions

This work provides new evidence on behavioural factors influencing dental antibiotic prescribing, as well as the resource constraints which affect the availability of certain antibiotics and diagnostic tests. Further research is required to fully understand their influence on use of antibiotics both as prophylaxis in dental procedures and for treatment of infections. This will inform the development of new approaches to optimising antibiotic use by dentists in Ghana and potentially in other low- and middle-income countries.

## Figures and Tables

**Table 1 antibiotics-11-01081-t001:** Specified behaviours using AACTT framework.

AACTT Framework	Dentistry Multi-Disciplinary Team	
**Actor**	Prescribers (Dentists, Dental surgery assistants, Physician assistants)	Dispensers (Pharmacists, Pharmacy technicians)
**Action**	Prescribe antibiotics to patients	Dispense antibiotic prescriptions to patients
**Context**	Dental clinics	Pharmacies
**Target**	Prudent and appropriate prescription of antibiotics	No prescribing target in hospital setting but role is to ensure good practice in dispensing antibiotic prescriptions
**Time**	When a patient attends the dental clinic to be treated for a dental condition	When a patient attends the pharmacy for antibiotics either via prescription or not

**Table 2 antibiotics-11-01081-t002:** Key themes and subthemes identified from thematic analysis.

Key Theme	Subthemes		
1. People and places	Prescriber and clinical context	Prescribing practices	Therapeutic use
			Prophylactic use
		Resource constraints	Access to antibiotics
			Diagnostic tests
			Infection prevention and control
		Influence of colleagues	Hierarchies
	Dispenser and pharmacies	Availability of antibiotics in the community	
	Patient and home environment	Delays in seeking professional help	
		Hygiene concerns	
		Medication adherence	
2. Knowledge and beliefs about AMR and ABS	Consequences of AMR		
	Reasons for AMR		
	Role of prescribers in tackling AMR		

**Table 3 antibiotics-11-01081-t003:** Interview participant quotes on ‘people and places’.

Theme or Subtheme	Quote and Participant Number
1.1 Prescribing practices	‘*P drug is the list of drugs that you have become comfortable giving. So this P drug, everybody has their own way of going about it. So probably drugs that your bosses have used before which they found to be very effective might become drugs that can form part of your P drugs*’. P4
1.1.1 Therapeutic use for treating active infections	‘*So we prescribe antibiotics where we are convinced or we are sure … that the conditions is mainly as a result of microbial contamination*’. P1
	‘*Sometimes you would give amoxicillin if the patient starts having very bad pain*’. P5
	‘*Well in this hospital, periodontitis is one of the common, common conditions for which antibiotics are written for in the dental unit*’. P2
1.1.2 Prophylactic use to prevent infections	‘*So most of the antibiotics we give are for prophylaxis against post-extraction complications and infection*’. P3
	‘*If you have a patient who has had endocarditis or somebody with heart valve or something, you have to give prophylactic antibiotics at least 30 min before you do any procedure that’s going to involve bleeding. So, like for such patients, even to do scaling and polishing, you would give antibiotics … to make sure that we don’t have any bacteria into the bloodstream*’. P5
1.2.1 Access to antibiotics—availability and affordability	‘*… if it’s available within the facility—that will also be a factor that would contribute to my prescribing drugs. Because if you would write an antibiotic and the patient would have to go round looking for it and they can’t find it, then what is the use? So, you write the readily available ones*’. P4
	‘*So our basic drugs are really available … unless we have Ludwig’s angina and stuff like that, our basics is penicillin and metronidazole*’. P5
	*‘… not all of them come here being insured under National Health Insurance Scheme (NHIS). If they are insured and they can get it under NHIS it’s good and I’ll prescribe that*’. P2
	*‘If the patient can seem to afford, we would write the more expensive ones, which we think will be effective. But if they, they can’t afford, we write the ones that they can afford for them and hope that it does the same work*’. P4
1.2.2 Diagnostic testing	*‘Where there is a need be for you know you’ve diagnosed an infection then I may give an antibiotic while waiting for my X-ray or whilst waiting for my culture and sensitivity results*’. P2
	*‘The only time we go beyond those guidelines is if we’ve done like I said, lab culture or analysis …*’. P1
1.2.3 Infection prevention and control	*‘Ideal situation is the mouth should be clean and the instruments are all brand new that you will use them, and sterilized*’. P2
	*‘In our setting, we prescribe antibiotics a lot and that’s quite unfortunate because we do not use new instruments for every patient. We have to autoclave and sterilize them. Elsewhere they have the luxury of discarding the instruments and using a new set of instruments … we even tag our environment as filthy*’. P3
1.3 Influence of colleagues—hierarchies	*If my boss is not around, I usually call to inform him before prescribing … describing the condition to him and all those things before prescribing*’. P8
	*‘… drugs that your bosses have used before which they found to be very effective might become drugs that can form part of your P drugs*’. P4
	*‘I don’t prescribe antibiotics in the hospital as a pharmacist because we have a dental unit which prescribes. I do recommend or prescribe maybe at home, over the weekends, where most dental facilities are closed, and then I make recommendations for suspected infections which I think will need antibiotic cover, like those with cavities and gum swelling*’. P7
	*‘They come around, do their presentations what not, but very often than not … it’s not like we are not open to new suggestions or new ideas or anything but per … I mean, we also do our own … it’s not just what they tell us, we don’t just take what they tell us at face value*’. P7
1.4 Availability of antibiotics without a prescription	*‘Availability per se may not influence whether you prescribe or not [in the hospital setting], but I think the general perception is that they’re accessible, they’re available, and there are so many pharmacies around. People can even walk into pharmacies for antibiotics and they can get it*’. P6
	*‘So the more rural you go, the less likely you’re going to have a hospital [and] dental care services [and] the more chemical shops you have … so people are likely to do self-medication … because the distance alone*’. P7
1.5 Delays in seeking professional help	*‘Culturally, people are used to taking antibiotics … abusing antibiotics. It’s a cultural problem*’. P7
	*‘They go to the drug store first, get antibiotics and will be taking and when they see there is no change, then they come … Usually they stay in the house for a very long time before the come*’. P9 *‘In our system it is more of a treat me when I’m sick … So they’re always in at the stage where there is the need to either extract it or do special procedures*’. P2
	*‘They will want you to give them antibiotics. And that also sometimes puts the pressure on you, the prescriber, in order to satisfy the patient’s demands*’. P6
1.6 Hygiene concerns	*‘… you can have a wound that is exposed to sand, and debris, gutter water, and whatever … You don’t expect us to attend to that wound and not give antibiotics*’. P1
	*‘It influences my decision … if the patient has a poor oral hygiene, the mouth itself is not too good, right for you to do any procedure*’. P2
1.7 Medication adherence	*‘Some of the patients you prescribe the medication for them and because they cannot afford, they are not able to buy all or take them*’. P3

**Table 4 antibiotics-11-01081-t004:** Interview participant quotes on ‘knowledge and training about AMR and AMS’.

Theme or Subtheme	Quote and Participant Number
2 Knowledge and training about AMR and AMS	‘*They came and trained my bosses some time ago, but I have, I have just read about it*’. P8
2.1 Consequences of AMR	‘*It’s a problem everywhere. Like, sooner or later we are going to have very serious issues with AMR, cause imagine you’re sick, we’ve identified certain organisms even like we’ve given you some medication but you’re not getting better because these organisms are having a field day … like it will be such a terrible thing and it will probably kill more than half of our population, if care is not taken*’. P1
	‘*For antibiotic stewardship, I know is to promote the use of antibiotics and to promote the appropriate use of drugs to prevent resistance. That’s what I know for antibiotic resistance stewardship, to do every possible best to use the right antibiotic to prevent the patient from being resistant to the drug*’. P12
2.2 Causes of AMR	‘*… some people refuse to follow the … way they are supposed to take the medicine … The person will take it, let’s say three days, and he’s finding out that he’s feeling better, so there is no need to continue. So they will leave the rest … they don’t complete the full course. So in that case, the person can develop resistance to the antibiotics*’. P11
	‘*I think the resistance is a main problem because it seems people are really misusing antibiotics*’. P11
	‘*So resistance can come from both ends. Either because the, the doctor is giving, going beyond the first line, to give third line and fourth line or the doctor is writing subtherapeutic dose so the dosage is lower than what can kill the bacteria or what can stop the activity. Or the patient discontinued the use of the antibiotic midway, before treatment, the period of treatment has elapsed. All these can cause antibiotic resistance*’. P4
	‘*So the problem is that most pharmacies are serving antibiotics without prescriptions*’. P3
2.3 Role of prescribers in tackling AMR	‘*One thing we tell them is never buy an antibiotic outside if … a doctor hasn’t seen, or a dentist hasn’t prescribed*’. P2

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
