# Peer review of "Exploring the Use of Antibiotics for Dental Patients in a Middle-Income Country: Interviews with Clinicians in Two Ghanaian Hospitals"

_antibiotics, 2022, doi:10.3390/antibiotics11081081_

Round 1

Reviewer 1 Report

This article raises issues that are very relevant and still a problem, especially in low-middle income countries. However, there are a few that I think need clarification from the author.

1. When I read the title, I thought it would display several LMICs countries... but it turned out to be only 1 country, which is Ghana. Shouldn't this LMICs be discussed in the introduction as a background to the problem?

2. In the method section, it is necessary to explain how the two hospitals were selected.

3. How long does this participant recruitment process take? need to be explained in the methods section.

4. Line 130...."...using the AACTT framework (see )" after "see" is there a missing word?

Author Response

This article raises issues that are very relevant and still a problem, especially in low-middle income countries. However, there are a few that I think need clarification from the author.

  1. When I read the title, I thought it would display several LMICs countries... but it turned out to be only 1 country, which is Ghana. Shouldn't this LMICs be discussed in the introduction as a background to the problem?

In the title we specified that our study was in Ghana, a middle-income country, but we have now added a sentence regarding provision of dental care in LMICs. (lines 56-58)

  1. In the method section, it is necessary to explain how the two hospitals were selected.

This is mentioned in the Introduction, but we have now also specified in the method how we selected the two hospitals. (lines 86-87)

  1. How long does this participant recruitment process take? need to be explained in the methods section.

We have added details about the time period for recruitment and for conducting the interviews. (lines 95 and 108)

  1. Line 130...."...using the AACTT framework (see )" after "see" is there a missing word?

Thank you, we have addressed this error by inserting ‘Table 1’.

Reviewer 2 Report

This study is based on subjective responses from providers and other staff involved in prescribing antibiotics for dental patients. Antibiotics resistance is a grave concern in the healthcare world, and therefore, research addressing this concern, like yours, is important. However, as the responses are based on subjective data, it may be important to conduct more studies with higher number of participants. 

Author Response

This study is based on subjective responses from providers and other staff involved in prescribing antibiotics for dental patients. Antibiotics resistance is a grave concern in the healthcare world, and therefore, research addressing this concern, like yours, is important. However, as the responses are based on subjective data, it may be important to conduct more studies with higher number of participants.

Thank you for your comments. We agree that further work is required to better understand the issues. This was a scoping study to test the methodology in this context and we plan to undertake a further study in Ghana and two other African countries to support development of an intervention to improve practice. This has been detailed in the final paragraph of the Discussion.

Reviewer 3 Report

Many thanks for the submission to this Journal: this paper is interesting with some additions and modifications.
1) in the introduction at line 52 add: In order to reduce the use of oral administration of antibiotics, the possibility of single dose local antibiotics or intravenous antibiotics has been studied"
add these citations in reference:

1. Busa A, Parrini S, Chisci G, Pozzi T, Burgassi S, Capuano A. Local versus systemic antibiotics effectiveness: a comparative study of postoperative oral disability in lower third molar surgery. J Craniofac Surg. 2014;25(2):708-9.

2. Reiland MD, Ettinger KS, Lohse CM, Viozzi CF. Does Administration of Oral Versus Intravenous Antibiotics for Third Molar Removal Have an Effect on the Incidence of Alveolar Osteitis or Postoperative Surgical Site Infections? J Oral Maxillofac Surg. 2017 Sep;75(9):1801-1808.

3. Chisci G, Capuano A, Parrini S. Alveolar Osteitis and Third Molar Pathologies. J Oral Maxillofac Surg. 2018 Feb;76(2):235-236.

2) The discussion section is correctly organized: it would be fantastic if you could fit a phrase regarding the antibiotics administration (therapy or phrophilaxis)

Author Response

Many thanks for the submission to this Journal: this paper is interesting with some additions and modifications.

1) in the introduction at line 52 add: In order to reduce the use of oral administration of antibiotics, the possibility of single dose local antibiotics or intravenous antibiotics has been studied"

add these citations in reference:

  1. Busa A, Parrini S, Chisci G, Pozzi T, Burgassi S, Capuano A. Local versus systemic antibiotics effectiveness: a comparative study of postoperative oral disability in lower third molar surgery. J Craniofac Surg. 2014;25(2):708-9.

  1. Reiland MD, Ettinger KS, Lohse CM, Viozzi CF. Does Administration of Oral Versus Intravenous Antibiotics for Third Molar Removal Have an Effect on the Incidence of Alveolar Osteitis or Postoperative Surgical Site Infections? J Oral Maxillofac Surg. 2017 Sep;75(9):1801-1808.

  1. Chisci G, Capuano A, Parrini S. Alveolar Osteitis and Third Molar Pathologies. J Oral Maxillofac Surg. 2018 Feb;76(2):235-236.

Thank you for your suggestion on adding this important point. However, on reading the suggested additional references we do not think they are relevant to our study since they are about a specific dental procedure and are not about studies conducted in a LMIC setting.

2) The discussion section is correctly organized: it would be fantastic if you could fit a phrase regarding the antibiotics administration (therapy or phrophilaxis)

We have highlighted that antibiotics are being overused for both treatment and prophylaxis in patients with dental conditions in the final sentence of the Discussion.